# Assessment of the Maximum Amount of Orthodontic Force for Dental Pulp and Apical Neuro-Vascular Bundle in Intact and Reduced Periodontium on Bicuspids (Part II)

**DOI:** 10.3390/ijerph20021179

**Published:** 2023-01-09

**Authors:** Radu Andrei Moga, Cristian Doru Olteanu, Mircea Botez, Stefan Marius Buru

**Affiliations:** 1Department of Cariology, Endodontics and Oral Pathology, School of Dental Medicine, Iuliu Hatieganu University of Medicine and Pharmacy, Str. Motilor 33, 400001 Cluj-Napoca, Romania; 2Department of Orthodontics, School of Dental Medicine, Iuliu Hatieganu University of Medicine and Pharmacy, Str. Avram Iancu 31, 400083 Cluj-Napoca, Romania; 3Department of Structural Mechanics, School of Civil Engineering, Technical University of Cluj-Napoca, Str. Memorandumului 28, 400114 Cluj-Napoca, Romania

**Keywords:** dental pulp, apical neuro-vascular bundle, periodontal-ligament breakdown, physiological hydrostatic-pressure, finite element analysis, orthodontic movements, maximum orthodontic force

## Abstract

This study examines 0.6 N–4.8 N as the maximum orthodontic force to be applied to dental pulp and apical NVB on intact and 1–8 mm reduced periodontal-ligament (PDL), in connection with movement and ischemic, necrotic and resorptive risk. In addition, it examines whether the Tresca finite-element-analysis (FEA) criterion is more adequate for the examination of dental pulp and its apical NVB. Eighty-one (nine patients, with nine models for each patient) anatomically correct models of the periodontium, with the second lower-premolar reconstructed with its apical NVB and dental pulp were assembled, based on X-ray CBCT (cone-beam-computed-tomography) examinations and subjected to 0.6 N, 1.2 N, 2.4 N and 4.8 N of intrusion, extrusion, translation, rotation, and tipping. The Tresca failure criterion was applied, and the shear stress was assessed. Forces of 0.6 N, 1.2 N, and 2.4 N had negligible effects on apical NVB and dental pulp up to 8 mm of periodontal breakdown. A force of 4.8 N was safely applied to apical NVB on the intact periodontium only. Rotation and tipping seemed to be the most invasive movements for the apical NVB. For the dental pulp, only the translation and rotation movements seemed to display a particular risk of ischemia, necrosis, and internal orthodontic-resorption for both coronal (0–8 mm of loss) and radicular pulp (4–8 mm of loss), despite the amount of stress being lower than the MHP. The Tresca failure criterion seems more suitable than other criteria for apical NVB and dental pulp.

## 1. Introduction

During orthodontic treatment, the dental pulp and apical neuro-vascular bundle (NVB) are, along with the periodontal ligament (PDL), the tissues that are subjected directly to orthodontic pressures and are prone to further damage: ischemia, necrosis, internal and external orthodontic root-resorption and further ligament loss [1,2]. If on an intact periodontium the PDL (with an average thickness of 0.2–0.3 mm) is capable of absorbing and dissipating the orthodontic forces up to a certain amount of force (which is still a debatable issue, with reports of both light and strong forces), this ability is seriously affected during periodontal resorption, leading to rapid further tissue-loss [1,2]. The circulatory system of both PDL, apical NVB and dental pulp are the first components to suffer from circulatory disturbances due to orthodontic pressures. The most sensitive component of PDL are the circulatory vessels found in the apical, middle, and cervical third, facing out and inwards among the various orientations’ collagen-fiber bundles [3]. The circulatory vessels insure the blood supply and the proper metabolism for the PDL, bone and dental pulp [3]. The apical third vessels are derived from the dental pulp, and include the apical NVB, while the middle third hold the perforating vessels, continuing with the gingival vessels from the cervical third [3]. The circulatory pressure in these PDL vessels was reported to be of 2–16 KPa (approximately 80% of the systolic pressure) [1,2,3,4,5,6,7,8]. If exceeded, it could produce circulatory disturbances leading to ischemia and followed by necrosis and resorptive processes [1,2,3,4,5,6,7,8].

From the biomechanical point of view, on an intact periodontium an optimal/maximal orthodontic force produces circulatory disturbances in the pulpal blood flow, the apical neuro-vascular bundle, and the remodeling of the periodontal ligament and bone, due to a change in tissue pressure that approximates the capillary vessel’s blood pressure (maximum physiological hydrostatic pressure/MHP) [1,8,9], which prevents vessel occlusion [10,11]. Dental pulp, which is extremely well innervated and vascularized, is anatomically interconnected with the periodontium via apical, lateral, and furcal foramina; thus, periodontal breakdown could inflict different levels of apical NVB and pulpal sufferance [3,12,13]. Still a sensitive issue, the amount of maximal orthodontic force differs for each tooth and patient [11]. Nevertheless, there is a high ability on the part of the periodontal tissues to adapt and to sustain damage without major further tissue loss, especially due to the various anatomical displays of the structures [3]. The apical NVB anatomical structure [with an average of 0.05–0.875 mm: 0.002–0.035 mm (40% of volume) for innervation and a similar diameter and volume percentage for the vascular component] is usually classified into three main categories (with various subvariants): thick nerve bundles alongside the blood vessels, thin spiral innervations, and neural bundles that surrounds the blood vessels [3]. The dental pulp has almost 40% of its structure as blood vessels and another 40% as nerve fibers, with an average diameter of the nerve bundles (a diameter also similar for the vascular component) in the middle third of the pulp of 0.007–0.012 mm, and 0.0005–0.06 mm in the furcation areas of the premolars [3]. Thus, due to anatomical differences, even on a healthy periodontium a higher force could produce blood-flow strangulation leading to inflammatory internal and external root-resorption, various levels of pulpal ischemic-sufferance and necrosis [10,11,14,15]. The selection of an optimal/maximal orthodontic force (agreed and considered to be light [16]) leads to the maximum rate of tooth movement, with speed adapted to particular clinical conditions, and producing minimal irreversible tissue damage, while the amount remains a subject of controversy [10,11,14,15,16]. Despite many in vivo and in vitro studies, on an intact periodontium, even the low-intensity stimulus applied for a prolonged time is sometimes reported to produce atrophic asymptomatic changes (e.g., pulp fibrosis reported for intrusion and extrusion) and external–internal orthodontic root-resorption [6,7,14,15]. Light forces up to 0.5–0.6 N seems safe for dental pulp and apical NVB on an intact periodontium [1,2,8,15], while 0.6–2.5 N may produce stresses exceeding the MHP, inducing ischemia and regressive changes [1,6,7,8,9,10,14,15]. Moreover, to further increase confusion on the subject of the maximal amount of force, recent reviews reported the poor quality of the in vivo studies observing that in an intact periodontium there is little evidence of pulp necrosis, emphasizing the need for more studies for a better understanding of the subject [11,14,17]. Still even less data is available regarding the maximal force safely applied on a reduced periodontium (having a higher susceptibility to pulp necrosis [18,19]), while no studies directly investigating the stress in the apical NVB, and dental pulp were found [2,8]. Due to the biomechanics in a reduced periodontium, the maximum amount of force is less, compared with the same force applied on the intact periodontium, while the ischemic, the orthodontic external and internal root-resorption and further periodontal-loss risks are even higher; thus, the physiological MHP should not be exceeded [2,8,13]. Recent studies (i.e., examining the relationship of MHP-stress during five common orthodontic movements, using Von Mises and Tresca criteria) reported 1.2 N maximal force for intact PDL and apical and the middle third of PDL with up to 8 mm of bone loss, and 0.2–0.4 N for the cervical third of PDL after 4 mm of tissue loss, while for dental pulp and apical NVB 0.2–0.6 N remains safe, irrespective of the degree of resorption [1,2,8,9]. However, no other reports about higher forces applied on apical NVB and dental pulp during a periodontal breakdown were found, except ours [1,2,8,9]. There are also clinical reports (including reviews [17]) of intrusive and extrusive lower forces 0.15–1.5 N applied with variable responses ranging from circulatory disturbances in the pulpal blood flow and histological changes to no significant differences [17,18,19,20,21]. The common conclusion of these studies was that, despite inconclusive evidence of pulpal circulatory disturbances and damages to the intact PDL, there is high risk of ischemia, necrosis and root resorption in the cases of variable degrees of periodontal injury [17,18,19,20,21], and thus the need to use small amounts of force is justified.

The study of stress distribution in PDL, dental pulp and apical NVB is granted only to in vitro simulations, due to complexity and the anatomically small size of each part, and the impossibility of individualized and particularized inquiry in vivo [1,2,4,5,6,7,8,13]. Thus, PDL and pulp studies used either anatomically idealized reconstructions or anatomical simplified CBCT (cone-beam-computed-tomography)-based 3D models, while no reconstructions of apical NVB were found [1,2,8,9]. The main issue consists in the small size of apical NVB (well under 1 mm), which is difficult to be identified and separated into the different shades of grey on the CBCT slice [1,2,8,14,15]. This process can usually be accurately carried out only by manual reconstruction performed by an experienced professional [1,2,8,9].

For a proper radiological examination, the CBCT must ensure a voxel size smaller than 0.2 mm (i.e., the average thickness of PDL), and be able to be used in daily clinical practice (e.g., minimal radiation dose, wide region of interest) [1,2,4,5,6,7,8]. Nevertheless for in vitro examinations, there are micro-CT with an extremely small voxel size, but which are not suited for human clinical-use, due to a high dose of radiation. The radiological data must be provided by daily clinical non-invasive X-ray examinations, to ensure the capture of distinct phases of periodontium under orthodontic treatment [1,2,8,10,11]. The collected data is reconstructed using an automated imaging software (i.e., grey-shades-algorithm-based) or manually [1,2,4,5,6,7,8]. Due to the complexity of the tissues, better anatomical accuracy is provided by the time-consuming manual reconstruction-process [1,2,8]. The reconstructions are then passed to FEA (finite-element-analysis) software allowing individual simulations and an analysis of the entire structure or limited to only one section or part [1,2,6,7,8].

The provided data is complex and complete, allowing a better understanding of the biomechanical behavior [1,2,4,5,6,7,8]. As a limit of this in vitro method, an FEA simulation cannot completely reproduce a clinical situation; nevertheless, it is the only available method for obtaining a detailed display of the biomechanical behavior of this type of tissues [1,2,4,5,6,7,8,13,14,15].

FEA is an extremely exact and widely used engineering method, but with limited use in dentistry, due to issues derived from the misunderstanding of the functioning principles which lead to reports contradicting in vivo studies [1,2,6,7,8,9,22,23,24,25,26,27,28]. These functioning principles are based on the yielding-of-materials theory (i.e., which type of stress deformation a material displays before its fracture or destruction) [2,29]. Thus, in order for an FEA to display correct results, the employed failure criteria must be specially designed for the type of material of the analyzed structure (i.e., brittle, ductile, or liquid/gas), and the correct anatomical information and physical properties must be carefully selected [1,2,8,9,29]. Moreover, to be indirectly confirmed, the quantitative reports must be corelated with the MHP [1,2,8,9,29]. There are multiple studies of the apical third of PDL employing an inappropriate failure criterion and no correlation with MHP-stress, reporting questionable results that tend to contradict in vivo reports [6,7,25,26,27,28]. Despite the fact that PDL, apical NVB and dental pulp were proved to resemble more ductile materials, numerous PDL studies employed: maximum principal S1 and minimum principal S3 (limited to brittle) [25,26,27,28,30] and hydrostatic pressure (limited to liquid-gas) [4,5,6,7,31,32] failure criteria, instead of using the adequate Von Mises (VM) and Tresca criteria, mathematically and physically designed for ductile materials [1,2,8,9,13,25,26,30,33]. As a consequence, they reported [6,7,25,26,27,28] quantitative values higher than MHP (which is viewed as an indirect validation tool) with no relationships with in vivo known clinical- data. Nevertheless, buy selecting the adequate failure criteria for ductile with the necessary relationships, recent studies [1,2,8,9] have reported quantitative results regarding the PDL, apical NVB and dental pulp, which match the in vivo results, proving the correctness of the method in dentistry, especially for tissues that cannot be studied otherwise.

The ductile field includes a large diversity of structures, ranging from steel to rubber, and with some of the anatomical tissues bearing most of the ductility features. In certain situations, such as those in living tissues, Tresca is more constraining and particularly more suited than Von Mises [2,8]. Thus, by using a potentially more exact criteria, this study examines the problem of the maximum amount of orthodontic force to be applied over the apical NVB and dental pulp in both the intact and reduced periodontium.

Thus, this study examines 0.6 N, 1.2 N, 2.4 N and 4.8 N as the maximum orthodontic force for dental pulp and apical NVB in intact and reduced PDL, in connection with each individual movement and ischemic, necrotic and resorptive risks. In addition, it examines whether the Tresca FEA criterion is more adequate for in vitro examination of dental pulp and apical NVB.

## 2. Materials and Methods

The simulation here is the result of a larger stepwise-progressive research study [1,2,8,9] (clinical protocol 158/02.04.2018) aiming to study the maximum amount of orthodontic force safely applied in an intact and reduced periodontium without any risks, and to identify the suitable FEA method/criteria to accurately perform this type of in vitro simulations.

Our study was performed with nine patients with reduced noninflamed periodontium (i.e., treated chronic-periodontitis stage II/III grade B periodontitis, enrolled in supportive therapy). This study initially considered more patients for inclusion, but only nine of them met the eligibility criteria: reduced noninflamed periodontium, complete mandibular arches with no teeth missing, various levels of bone height, indication of orthodontic treatment and availability for follow-up through treatment. The mandibular area (including first and second molar and the two premolars) was investigated using 0.075 mm voxel-size CBCT (ProMax 3DS, Planmeca, FI-00880 Helsinki, Finland).

Based on the CBCT data, anatomically correct reconstructions of the structural tissues (i.e., enamel, dentin, dental pulp and its NVB, PDL, cortical and trabecular bone) have been conducted, employing the manual image-segmentation technique using AMIRA 5.4.0 software (AMIRA, version 5.4.0, Visage Imaging Inc. 300 Brickstone Square, Suite 201 Andover, MA 01810, USA), based on the Hounsfield grey-shades units and conducted by a single experienced clinician. Based on the different grey shades, the anatomical components were identified and selected individually from each of the DICOM slices of the CBCT. All identified components were then assembled into a single 3D mesh model (i.e., one for each of the nine patients). Thus, nine models having the second premolar (a single and two rooted, with different anatomical shapes) and various levels of periodontal breakdown, limited mostly to the cervical third of PDL, were obtained. The manual image-segmentation technique (complex and difficult) was preferred for its anatomical accuracy (the automated reconstruction-algorithm refines by simplification, committing errors when showing extremely small tissues and very similar grey shades). In each of the nine models the molars and first premolar were replaced by cortical and trabecular bone, while the missing bone and PDL were reconstructed as accurately as possible, in order to obtain nine models with intact periodontium (Figure 1). In each model the dentin replaced cementum (due to the similarity of its mechanical properties, Table 1), and PDL had a variable average thickness of 0.15–0.225 mm, and was reconstructed with its apical NVB. Each of the nine intact-periodontium models was the subject of a manual gradual horizontal periodontal-breakdown of 1 mm in height (up to 8 mm of loss), obtaining a total number of eighty-one analyzed models (nine models/patient) with the 2nd lower premolar.

The intact-periodontium mesh models had 5.06–6.05 million C3D4 tetrahedral elements, 0.96–1.07 million nodes, and a global element-size of 0.08–0.116 mm (extremely fine-grained mesh subjected to mesh-convergence testing). The manual reconstruction technique implied also a limited number of surface anomalies and irregularities naturally present in all models, but with quasi-continuity in stressed areas; nonetheless, the internal algorithm-based control processes were passed.

FEA simulation was performed by employing the Tresca failure criterion (shear stress) and using ABAQUS 6.13-1 software (Dassault Systèmes Simulia Corp., Stationsplein 8-K, 6221 BT Maastricht, The Netherlands). The orthodontic forces applied over the bracket were 0.6 N (approx. 60 g), 1.2 N (approx. 120 g), 2.4 N (approx. 240 g) and 4.8 N (approx. 480 g), individually for intrusion, extrusion, translation, rotation, and tipping movements (Figure 1), to reproduce as accurately as possible the effects of the movements.

The physical properties of tissues (the boundary conditions, Table 1) were homogeneity, isotropy, linear elasticity, and perfectly bonded interfaces, with no displacements of the base of the models. All models were subjected to similar boundary conditions, material properties and loading conditions.

Tresca shear-stresses were found and were displayed qualitatively (i.e., numerically expressed as a color-coded projection) for the dental pulp and its NVB in Figure 2 and Figure 3 (for one of the nine models), while quantitatively (as an average, for all models) in Table 2, Table 3, Table 4 and Table 5. The average quantitative values were each time referred to the 16 KPa of physiological MHP, and the risks of external–internal orthodontic coronal- and radicular-resorption, ischemia-necrosis, and further periodontal loss were individually examined. Stress-increase speed was correlated with quantitative stress values for the intact periodontium, as a reference point. The displayed quantitative (Table 2, Table 3, Table 4 and Table 5) and qualitative results were also associated with our earlier observations [1,2,8] employing the Tresca and Von Mises failure-criteria for dental pulp and its NVB.

## 3. Results

The FEA analysis here was performed with nine patients (mean age 29.81 ± 1.45 years, four males, informed oral consent) with reduced noninflamed periodontium, and over a total of eighty-one 3D models (nine models/patient).

The most significant stresses were found in the apical NVB for all five movements and forces, for both intact and reduced periodontium (Figure 2 and Figure 3, red and orange color-coded projection, and Table 2, Table 3, Table 4 and Table 5). The apical NVB stress was quantitatively higher than the pulpal stress in all simulations. Qualitatively, the stressed area displayed the same extent of the color-coded distribution for all four forces, with the only difference being quantitative higher values (Figure 2 and Figure 3). No visible influence of age, gender or periodontal status was observed.

In the apical NVB in both intact and reduced periodontium, 0.6 N and 1.2 N displayed for all five movements quantitative values under the 16 KPa of the physiological MHP (Table 2 and Table 3). A higher force of 2.4 N produced higher stresses than the MHP value at 8 mm of tissue loss only (except for rotation, with higher stresses after 5 mm of bone loss) (Table 4). A force of 4.8 N was safely applied only on the intact periodontium and up to 1–2 mm of resorption (Table 5). Rotation and tipping seem to be the most invasive movements for the apical NVB, closely followed by intrusion and extrusion, with the least being translation. Thus, 2.4 N orthodontic force seems to be safe for the apical NVB, without any significant ischemic and resorptive risks. However, limited localized areas of higher stress (yellow, orange, and red color-coded projections) are displayed by the apical NVB in each of the five movements, suggesting small areas of ischemia and potential external–internal apical root-resorption (Figure 2 and Figure 3A–E).

The dental pulp, despite displaying a significantly lower amount of stress than its apical NVB, seemed to be more sensitive to rotation and translation. Thus, the coronal pulp showed visible-stress areas (various color-coded shades of blue) for translation and rotation movements (Figure 2 and Figure 3C,D), especially for translation, and up to 8 mm of resorption (with a maximum of visible stress from 0–4 mm of tissue loss), suggesting limited ischemia and potential coronal internal-orthodontic-resorption. The potential internal root-resorption process was seen in the radicular pulp displaying significant visible-limited-stress areas for the same translative movement from 4 mm to 8 mm of bone loss, with a maximum at the highest rate of resorption (Figure 2 and Figure 3C). However, up to 4.8 N of applied force, all the quantitative stresses were under the MHP physiological limit. The circulatory disturbances that could potentially lead to various degrees of ischemia-necrosis and orthodontic internal coronal- and radicular-resorption seem to appear for rotational and translational movements. The translation movement seems particularly to display a more advanced ischemic and resorptive risk for both coronal (0–8 mm of tissue loss) and radicular pulp (4–8 mm of tissue loss).

Thus, based on the herein quantitative results here, 0.6 N, 1.2 N and 2.4 N of orthodontic force seems free of any significant and major ischemic-necrotic and internal–external resorptive risks for apical NVB and dental pulp, in both the intact and the reduced periodontium. However, limited areas of potential internal and external orthodontic-resorption are visible for both apical NVB and dental pulp, with the translational movement displaying the most visible internal coronal- and radicular-resorptive risks.

All five movements displayed a directly proportional correlation between the shear-stress increase and the progress of periodontal breakdown, up to a doubling of the quantitative amount of shear stress. When compared with the Von Mises failure-criterion of earlier reports, the quantitative results were found to be within the recognized limits of approx. 30% higher (i.e., 34.5% for the present results).

To achieve a suitable interpretation of these results and a clear image of the biomechanical behavior of tissues (i.e., PDL, apical NVB and dental pulp working together under orthodontic forces), associations and relationships with the results of the same simulation conducted in an earlier report on PDL were also performed. As a result, PDL was found to have significantly higher ischemic-necrotic and resorptive risks than the apical NVB, while the dental pulp was shown to have the least.

## 4. Discussion

This study of eighty-one 3D models (nine patients, nine models/patient) examined whether a 0.6 N, 1.2 N, 2.4 N and 4.8 N orthodontic force could be considered maximal and safe for apical NVB and dental pulp for several orthodontic movements in a periodontal-breakdown process. Relationships among MHP for each individual movement, ischemic, necrotic, internal-external orthodontic-resorptive risks and the PDL (tissue holding the apical NVB) were examined. In addition, an inquiry into Tresca criterion as more suitable for dental pulp and its apical NVB was made. It must be acknowledged that for this type of investigation only in vitro FEA methods are possible. Moreover, few data are available on this issue, due to technical difficulties related to the complicated anatomy of the region and the difficulty of the isolation and reconstruction of tissue from the CBCT slices [1,2,8].

In order to have a clear image of the biomechanical behavior of tissues (i.e., PDL, apical NVB and pulp) working together under orthodontic forces, and for the indirect validation process, associations and relationships with previous PDL reports [2,8,9] were performed.

The numerical results have shown that 0.6 N, 1.2 N and 2.4 N are safe to be used for both apical NVB and dental pulp up to 8 mm of periodontal breakdown, displaying amounts of stress lower than MHP (Table 2, Table 3 and Table 4), with no circulatory or resorptive risks, in agreement with previous in vitro [1,2,8,9,10,11,12,13,14,15,16], in vivo [17,18,19,20,21], and Proffit et al. [16] reports. However, a higher force of 4.8 N can be safely applied only on an intact periodontium (Table 5). The pulpal stress was significantly lower than its apical NVB stress, suggesting that orthodontic forces have the highest impact on the apical NVB (found in the apical third of the PDL), while dental pulp remains mostly untouched (Figure 2 and Figure 3, Table 2, Table 3, Table 4 and Table 5).

The rotation and tipping forces seem to be the most invasive movements for the apical NVB, causing limited ischemic and resorptive risks, contrary to Minch et al. [13], who reported intrusion to be the most invasive and causing risks. Nevertheless, small, localized areas of higher stress (color coded in yellow, orange, and red) were displayed by the apical NVB in each of the five movements, suggesting limited areas of ischemia and potential external—internal apical orthodontic root-resorption that might occur, in agreement with various clinical reports [17,18,19,20,21].

Dental pulp is more sensitive to the rotation and translation movements (Figure 2 and Figure 3C,D), displaying visible signs (color-coded in blue) of higher-stress areas in coronal and radicular pulp. Thus, despite showing amounts of stress lower than MHP, visible localized areas of potential ischemia, necrosis, and internal orthodontic coronal-resorption are displayed in the coronal pulp, especially for translation (0–8 mm bone loss) and rotation (up to 4 mm loss). The potential internal radicular root-resorption areas are visible only for the translation movement, seemingly stimulated by the bone-loss process (Figure 2 and Figure 3C).

These results are in line with earlier in vitro observations [1] (employing VM and Tresca criteria in apical NVB and pulp, using 0.2–0.6 N), Profitt et al. [16], and in vivo reports of dental-pulp behavior [17,18,19,20,21] under 0.15–1.5 N of applied force.

However, because of limited data on the biomechanical behavior of the apical NVB and dental pulp, the anatomical topography in the apical PDL, the PDL-pulp circulatory anastomosis, the need for result validation, and a better understanding of the behavior, an association with the quantitative reports regarding PDL was needed and found suitable.

For better accuracy of the correlation/association, PDL should be reconstructed with the apical NVB. Few studies [2,8] met this requirement, reporting 0.6–1.2 N to be safely applied in the intact and reduced periodontium, displaying quantitative stress values lower than MHP and almost no ischemic or resorptive risks for the apical and middle-third PDL. However, 1.2 N was reported to display some limited areas with ischemic-resorptive risks in the 1 mm of height of the cervical PDL [2,8]. Thus, the most sensitive part of the functional complex PDL-apical NVB-dental pulp seems to be PDL (i.e., especially the first mm of height of the cervical third) and not the apical NVB. Consequently, the maximal force should be around 1.2 N (120 g) to avoid any risks, in accordance with both in vivo [17,18,19,20,21] and in vitro [1,2,8,9,10,11,12,13,14,15,16] reports, and Proffit et al. [16].

It must be acknowledged as a limitation that the simulations here were performed with pure orthodontic forces, while in clinical situations (as in those reported in vivo [17,18,19,20,21]) there is an association of them that could lead to the observation that clinically the quantitative stresses might be even smaller than those here.

Our approach to the use of around 1 N (up to 120 g) of appliance as the maximal orthodontic force in periodontal breakdown for apical NVB, dental pulp and PDL is in agreement with various multiple studies [1,9,11,13,14,16,18,32,34,35,36,37,38], but contradicts other studies [4,5,6,7,32].

In disagreement with the results here, Hohmann et al. [6,7] reported extensive apical external-root-resorption for 0.5–1 N of intrusion (with stress values of approx. 80 KPa in the apical third of PDL) and 3–6 N lingual torque (with stress values of approx. 38.8–40.4 KPa in the apical third PDL) for two rotting upper premolars (reconstructed using AMIRA software), employing the ABAQUS software and using the hydrostatic-pressure criteria. However, in a previous report [2], the hydrostatic-pressure criteria was proven to be unsuited for the study of PDL.

Contrary to our simulation, Wu et al. [4,5,32], by also employing hydrostatic-pressure criteria in an intact periodontium, reported various optimal forces for intact PDL (with a range of 0.28–3.31 N) for canine, premolar, and lateral incisive, with significant differences for the same tooth (e.g., canine: rotation 1.7–2.1 N [32] and 3.31 N [5]; extrusion 0.38–0.4 N [32] and 2.3–2.6 N [4]; premolar: rotation 2.8–2.9 N [5]), much higher than the 0.6–1.2 N report here, Proffit et al. [16] (0.1–1 N), Hemanth et al. [23,24] (0.3–1 N), and other in vivo [17,18,19,20,21] and in vitro [1,2,8,9,10,11,12,13,14,15,16] studies. Moreover, both Javed et al. [17] and Bauss et al. [18,19,20,21] in clinical reports of dental pulp behavior under orthodontic treatment, reported 0.15–0.2 N (15–20 g) [18,19,20,21] and 0.15–1.5 N [17] of intrusion and extrusion to be safe for the dental pulp and free of ischemic risks, unless there were previous varying levels of periodontal injuries and trauma (where the risks were found to be significant), in agreement with here and previous reports [1,2,8].

For a simpler clarification of the problem of the accuracy of FEA analyses, the basic principle is to use a criterion adequate to the type of investigated material (i.e., brittle, or ductile for the dental tissues). The division of materials into these two categories is based on the theory of yielding of materials (i.e., the type of permanent/reversible plastic or elastic deformation displayed by the stressed material before its fracture or destruction) [1,2,8,9,29].

Ductile material displays a certain amount of reversible deformation before destruction, while brittle displays a permanent deformation. The living tissues have more ductile or brittle resemblance, without having all their features. Thus, PDL, dental pulp and its apical NVB display more ductile resemblance but also with a limited brittle flow. For ductile materials, Tresca and Von Mises are the adequate criteria, with the Tresca more suitable for materials displaying a polymorphic (various combinations of brittle–ductile features, 3D non-smooth behavior, and a narrower elastic-region) behavior under stress [2,8].

Thus, when interpreting the FEA studies, the suitable failure criteria adequate for the material type is a mandatory topic of discussion (aiming to obtain exact results, as in the engineering field). If the non-adequate criteria are used the results suffer, with various degrees of alteration [2,8,29]. There are multiple FEA studies of the periodontium employing the hydrostatic-pressure criteria (adequate only for liquids or gas, and where there are no shear stresses) analyzing the PDL and tooth, and supplying quantitative results contradicting clinical data [4,5,6,7,32]. Other studies employ brittle-material criteria (maximum S1, minimum S3 criterial) in the study of PDL (which is a ductile-like material), displaying quantitative stress values exceeding the MHP [23,24,26,27,28,30]. Therefore, despite numerous reports [6,7,23,24,26,27,28,30] of apical-third PDL stress, employing S1/S3 and hydrostatic-pressure failure criteria, a comparation with their results is not acceptable, due to major differences in employment indications (e.g., type of material and boundary conditions).

Only one Tresca report [2] (with apical NVB reconstruction) and a few Von Mises studies [25,26,30,33,39] assessing the apical third of PDL (but without an apical NVB reconstruction) were found reporting quantitative results. Results here are in line with Gupta et al. [39] (VM, single model, intact periodontium, apical PDL without apical NVB and pulp, upper incisor, 30452 elements, 61900 nodes, 0.3 N, extrusion 0.07 KPa, tipping 0.26 KPa). Higher quantitative values had been reported by Show et al. [25] (unidentified criteria–VM/S1, single model, intact periodontium without apical NVB, upper incisor, 11924 elements, 20582 nodes, apical stress, intrusion/extrusion: pulp 5.42 KPa, PDL 2 KPa; tipping: pulp 3.28–4.88 KPa, PDL 0.89–1 KPa, rotation: pulp 0.68 KPa, PDL 0.02 KPa), Roscoe et al. [30] (VM, single model, intact periodontium, premolar, without apical NVB, 1.67 mil elements, 0.25 N, intrusion 1.1 KPa, tipping 2.9 KPa), Toms et al. [26] (VM, single model, intact periodontium, lower premolar, 1674 elements, 5205 nodes, apical PDL, 1 N of extrusion, 8 KPa), and Merdji et al. [33] (VM, single model, intact periodontium, lower 1st molar, 557974 elements, apical third PDL, 10 N of intrusion 29.48 KPa; 3 N of tipping 8.96 KPa; 3 N of translation 6.78 KPa). These differences could be due to the tooth models, anatomical-accuracy modelling (a reduced number of elements and nodes) and applied force (boundary conditions). The Tresca report [2] (0.5 N, 2nd lower premolar with apical NVB, 0–8 mm bone loss, intrusion/extrusion: 2.5 KPa, translation: 1.68 KPa, rotation: 1.94 KPa, tipping: 1.29 KPa for the apical third PDL) is in line with the quantitative results here.

After the adequate failure-criteria is selected, the reconstruction of the anatomically correct 3D-model of the structure, enforced with the physical properties/boundary conditions, should follow. These two requirements are mandatory for accurate quantitative results. Usually in FEA analysis (here included) the structures’ properties include isotropy, linear elasticity, and homogeneity, which are correct from the biomechanical point of view for extremely small forces [1,2,8,9,29].

The living structures also display a non-smooth behavior [1,2,8,9,29]. Hemanth et al. [23,24], in a simulation of linearity vs. non-linearity with 0.3–1 N (30–100 g) of intrusion and lingual torque, in an idealized anatomy of the maxillary incisive periodontal-ligament, employing maximum S1 and minimum S3 principal stress (i.e., unsuitable for the PDL study, due to the criteria’s mathematical algorithm being designed only for brittle structures [1,2,8,9,29]), reported 20–50% less quantitative applied force needed for non-linearity compared to linearity. Despite the obvious problem regarding the selection of failure criteria, these studies [23,24] suggested that a lower force of 0.3–1 N could produce higher amounts of stress in clinical conditions (meaning a faster surpassing of the physiological values of MHP), leading to ischemia, necrosis and resorptive risks, even at such a lower stress, in total contradiction to clinical reports [17,18,19,20,21] regarding dental pulp, other in vitro studies [1,9,11,13,14,16,18,32,34,35,36,37,38] and Proffit et al. [16].

The number of studied models and their anatomical accuracy is highly dependent on the type and number of elements and nodes (a higher number usually means more anatomical details), and if not respected could alter the quantitative results. An anatomically exact model should be reconstructed based on CBCT radiological examinations, (reflecting particularities and individualities) and not on an idealized anatomically simplified model. Here, we employed eighty-one models of the second lower-premolar with up to 40–12,731 times more C3D4 tetrahedral elements and up to 4.4–1463 times more nodes than in previous studies [6,7,17,18,19,20,21,23,24,25,26,27,28,30,33,39].

Clinically, there is an association and combination of movements, and thus the quantitative stresses that are displayed in the apical NVB, PDL and dental pulp are expected to be lower than those here. It must be acknowledged as a limitation of this method (in addition to those mentioned earlier throughout this study), that the FEA is a mathematically based algorithmic method that analysis a clinically extremely complex biomechanical-behavior that cannot be entirely reproduced in vitro. Nevertheless, there are no other in vivo or in vitro methods available to investigate these small structures, and thus FEA, despite all its limitations, remains the best and only available tool. To gain a better knowledge of the correlations and interrelations between the internal and external orthodontic coronal and radicular resorption, the pulp-PDL-apical NVB complex biomechanical behavior, the amount of applied force and the periodontal breakdown, more FEA simulations employing suitable failure criteria are needed.

## 5. Conclusions

Based on the simulations here, and taking into account the methodological advantages and limitations, some conclusions can still be drawn:
A continuous orthodontic force of 0.6 N, 1.2 N, and 2.4 N has negligible effect on NVB and even lesser on dental pulp with up to 8 mm of periodontal breakdown, with rotation and tipping seeming to be the most invasive. However, 4.8 N of force was safely applied for apical NVB only in the intact periodontium.Localized areas of higher stress (color-coded in yellow, orange, and red) are displayed by the apical NVB in each of the five movements, suggesting small areas of ischemia and external–internal apical orthodontic root-resorption.The dental pulp seems to be more sensitive to the rotation and translation movements. Translation and rotation movements seem to display a particular risk of localized ischemia, necrosis, and potential internal orthodontic-resorption for both coronal- (0–8 mm of loss) and radicular-pulp (4–8 mm of loss), despite the amount of stress being lower than MHP.The Tresca failure criteria seem more suited to the study of apical NVB and dental pulp. To have a clear image of the biomechanical behavior of tissues working together under orthodontic forces, associations and relationships with PDL must be examined.

## 6. Practitioner Points

As clinical reports concluded that a previously traumatized tooth with various degrees of periodontal injury is more sensitive to ischemia, necrosis and orthodontic coronal and radicular resorption during orthodontic treatment, lower applied-forces should be used; 0.6 N, 1.2 N, and 2.4 N had a negligible effect on NVB, and even less on dental pulp with up to 8 mm of periodontal breakdown. Thus, ischemia, pulpal necrosis and internal and external resorption risks due to orthodontic movements, are extremely low. However, for the translational and rotational movements, particular attention should be paid, due to visible signs of higher stress displayed in the coronal and radicular pulp during periodontal breakdown. Due to close relationships between the PDL and apical NVB subjected together to orthodontic forces, the amount of maximal orthodontic force should consider more the PDL resistance to ischemia (higher risks of ischemia and further periodontal loss if subjected to more than 1.2 N of force) than that of the NVB or dental pulp (little or no risk of ischemia).

## Figures and Tables

**Figure 1 ijerph-20-01179-f001:**
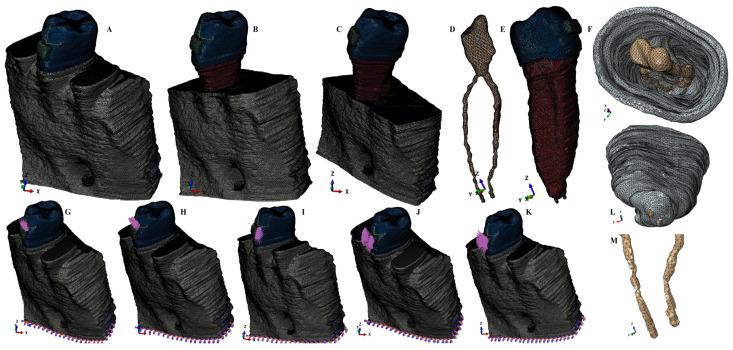
Mesh model: (**A**) 2nd lower-right premolar model, with intact periodontium, (**B**) 4 mm bone loss, (**C**) 8 mm bone loss, (**D**) dental pulp with apical NVB, distal-vestibular view, (**E**) 2nd premolar with bracket, 4 mm bone loss, (**F**) PDL and dental pulp, cervical-coronal view; applied load vectors: (**G**) intrusion, (**H**) extrusion, (**I**) translation, (**J**) rotation, (**K**) tipping, (**L**) PDL with apical NVB, apical view, (**M**) apical NVB and radicular pulp, apical third.

**Figure 2 ijerph-20-01179-f002:**
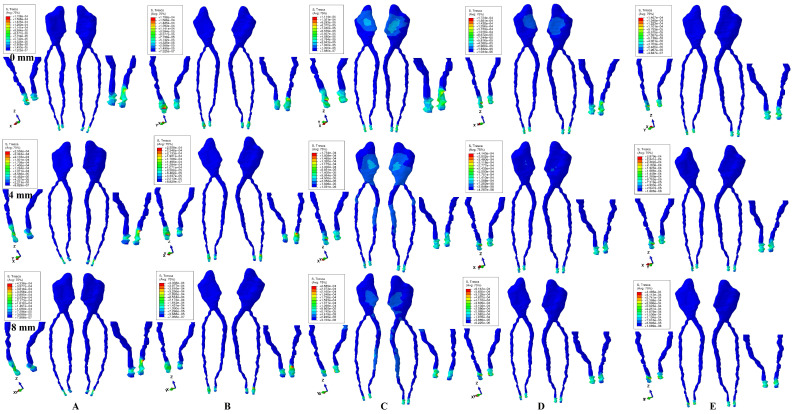
Tresca shear-stress display in dental pulp with its apical NVB (distal-vestibular and mesial-lingual, global, and detailed views) for 0.6 N (the quantitative values are in MPa) (intact, 4 mm and 8 mm reduced periodontium): (**A**) intrusion, (**B**) extrusion, (**C**) translation, (**D**) rotation, (**E**) tipping; the highest stressed areas are color-coded in red and orange.

**Figure 3 ijerph-20-01179-f003:**
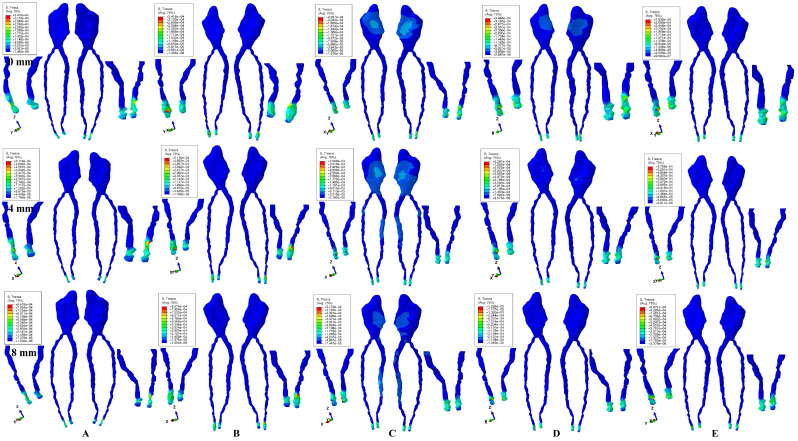
Tresca shear-stress display in dental pulp with its apical NVB (distal-vestibular and mesial-lingual, global, and detailed views) for 1.2 N (the quantitative values are in MPa) (intact, 4 mm and 8 mm reduced periodontium): (**A**) intrusion, (**B**) extrusion, (**C**) translation, (**D**) rotation, (**E**) tipping; the highest stressed areas are color-coded in red and orange.

**Table 1 ijerph-20-01179-t001:** Elastic properties/physical properties of materials in GPa.

Material	Young’s Modulus, E (GPa)	Poisson Ratio, ʋ	Refs.
Enamel	80	0.33	[1,2,8,9]
Dentin/Cementum	18.6	0.31	[1,2,8,9]
Dental Pulp/Apical-NVB	0.0021	0.45	[1,2,8,9]
PDL	0.0667	0.49	[1,2,8,9]
Cortical bone	14.5	0.323	[1,2,8,9]
Trabecular bone	1.37	0.3	[1,2,8,9]
Bracket (Cr-Co)	218	0.33	[1,2,8,9]

**Table 2 ijerph-20-01179-t002:** The Tresca shear-stress average values (KPa) produced by 0.6 N of orthodontic forces.

Resorption (mm)		0	1	2	3	4	5	6	7	8
Intrusion 0.6 N	NVB	**1.71**	**1.92**	**2.14**	**2.35**	**2.56**	**3.01**	**3.45**	**3.90**	**4.34**
	% NVB	1.00	1.13	1.25	1.38	1.50	1.76	2.02	2.28	2.54
	a	**0.15**	**0.17**	**0.19**	**0.20**	**0.22**	**0.26**	**0.30**	**0.33**	**0.37**
	% a	1.00	1.12	1.24	1.36	1.48	1.73	1.98	2.23	2.47
	c	**0.15**	**0.17**	**0.19**	**0.20**	**0.22**	**0.26**	**0.30**	**0.33**	**0.37**
	% c	1.00	1.12	1.24	1.36	1.48	1.73	1.98	2.23	2.47
Extrusion 0.6 N	NVB	**1.71**	**1.92**	**2.14**	**2.35**	**2.56**	**3.01**	**3.45**	**3.90**	**4.34**
	% NVB	1.00	1.13	1.25	1.38	1.50	1.76	2.02	2.28	2.54
	a	**0.15**	**0.17**	**0.19**	**0.20**	**0.22**	**0.26**	**0.30**	**0.33**	**0.37**
	% a	1.00	1.12	1.24	1.36	1.48	1.73	1.98	2.23	2.47
	c	**0.15**	**0.17**	**0.19**	**0.20**	**0.22**	**0.26**	**0.30**	**0.33**	**0.37**
	% c	1.00	1.12	1.24	1.36	1.48	1.73	1.98	2.23	2.47
Translation 0.6 N	NVB	**1.11**	**1.27**	**1.43**	**1.59**	**1.75**	**1.97**	**2.18**	**2.38**	**2.59**
	% NVB	1.00	1.14	1.28	1.43	1.58	1.77	1.95	2.14	2.32
	a	**0.11**	**0.12**	**0.13**	**0.15**	**0.16**	**0.18**	**0.21**	**0.23**	**0.25**
	% a	1.00	1.13	1.25	1.36	1.49	1.69	1.90	2.11	2.31
	c	**0.28**	**0.32**	**0.37**	**0.41**	**0.45**	**0.51**	**0.56**	**0.62**	**0.67**
	% c	1.00	1.13	1.28	1.43	1.58	1.78	1.97	2.16	2.37
Rotation 0.6 N	NVB	**1.72**	**2.33**	**2.93**	**3.54**	**4.14**	**4.64**	**5.14**	**5.64**	**6.14**
	% NVB	1.00	1.35	1.70	2.05	2.40	2.69	2.98	3.27	3.56
	a	**0.15**	**0.21**	**0.27**	**0.33**	**0.38**	**0.43**	**0.48**	**0.52**	**0.57**
	% a	1.00	1.35	1.73	2.14	2.51	2.79	3.10	3.41	3.71
	c	**0.29**	**0.43**	**0.51**	**0.62**	**0.73**	**0.82**	**0.91**	**0.99**	**1.08**
	% c	1.00	1.48	1.76	2.15	2.50	2.82	3.12	3.42	3.71
Tipping 0.6 N	NVB	**1.47**	**1.82**	**2.18**	**2.53**	**2.88**	**3.28**	**3.69**	**4.09**	**4.49**
	% NVB	1.00	1.24	1.48	1.72	1.96	2.24	2.51	2.79	3.06
	a	**0.15**	**0.18**	**0.21**	**0.23**	**0.25**	**0.29**	**0.33**	**0.36**	**0.39**
	% a	1.00	1.16	1.37	1.49	1.69	1.89	2.16	2.36	2.58
	c	**0.15**	**0.18**	**0.21**	**0.23**	**0.25**	**0.29**	**0.33**	**0.36**	**0.39**
	% c	1.00	1.16	1.37	1.49	1.69	1.89	2.16	2.36	2.58

NVB apical stress, a: apical-third pulpal stress, c: coronal pulpal stress, % NVB: no. of times of stress increase, % a: no. of times of apical-third pulpal-stress increase, % c: no. of times of coronal-stress increase.

**Table 3 ijerph-20-01179-t003:** The Tresca shear-stress average values (KPa) produced by 1.2 N of orthodontic force.

Resorption (mm)		0	1	2	3	4	5	6	7	8
Intrusion 1.2 N	NVB	**3.42**	**3.85**	**4.28**	**4.70**	**5.12**	**6.01**	**6.90**	**7.79**	**8.68**
	% NVB	1.00	1.13	1.25	1.37	1.50	1.76	2.02	2.28	2.54
	a	**0.30**	**0.34**	**0.37**	**0.41**	**0.44**	**0.52**	**0.59**	**0.67**	**0.74**
	% a	1.00	1.13	1.23	1.37	1.47	1.73	1.97	2.23	2.47
	c	**0.30**	**0.34**	**0.37**	**0.41**	**0.44**	**0.52**	**0.59**	**0.67**	**0.74**
	% c	1.00	1.13	1.23	1.37	1.47	1.73	1.97	2.23	2.47
Extrusion 1.2 N	NVB	**3.42**	**3.85**	**4.28**	**4.70**	**5.12**	**6.01**	**6.90**	**7.79**	**8.68**
	% NVB	1.00	1.13	1.25	1.37	1.50	1.76	2.02	2.28	2.54
	a	**0.30**	**0.34**	**0.37**	**0.41**	**0.44**	**0.52**	**0.59**	**0.67**	**0.74**
	% a	1.00	1.13	1.23	1.37	1.47	1.73	1.97	2.23	2.47
	c	**0.30**	**0.34**	**0.37**	**0.41**	**0.44**	**0.52**	**0.59**	**0.67**	**0.74**
	% c	1.00	1.13	1.23	1.37	1.47	1.73	1.97	2.23	2.47
Translation 1.2 N	NVB	**2.23**	**2.54**	**2.86**	**3.18**	**3.51**	**3.94**	**4.35**	**4.76**	**5.17**
	% NVB	1.00	1.14	1.28	1.43	1.57	1.77	1.95	2.13	2.32
	a	**0.22**	**0.25**	**0.27**	**0.29**	**0.32**	**0.37**	**0.41**	**0.46**	**0.50**
	% a	1.00	1.14	1.23	1.32	1.45	1.68	1.86	2.09	2.27
	c	**0.57**	**0.65**	**0.73**	**0.82**	**0.90**	**1.01**	**1.12**	**1.23**	**1.35**
	% c	1.00	1.14	1.28	1.44	1.58	1.77	1.96	2.16	2.37
Rotation 1.2 N	NVB	**3.45**	**4.65**	**5.86**	**7.07**	**8.29**	**9.29**	**10.29**	**11.28**	**12.29**
	% NVB	1.00	1.35	1.70	2.05	2.40	2.69	2.98	3.27	3.56
	a	**0.31**	**0.42**	**0.53**	**0.66**	**0.77**	**0.86**	**0.95**	**1.05**	**1.14**
	% a	1.00	1.35	1.71	2.13	2.48	2.77	3.06	3.39	3.68
	c	**0.58**	**0.86**	**1.02**	**1.25**	**1.45**	**1.64**	**1.81**	**1.99**	**2.15**
	% c	1.00	1.48	1.76	2.16	2.50	2.83	3.12	3.43	3.71
Tipping 1.2 N	NVB	**2.93**	**3.65**	**4.35**	**5.06**	**5.76**	**6.57**	**7.37**	**8.18**	**8.97**
	% NVB	1.00	1.25	1.48	1.73	1.97	2.24	2.52	2.79	3.06
	a	**0.25**	**0.33**	**0.39**	**0.45**	**0.51**	**0.57**	**0.65**	**0.71**	**0.78**
	% a	1.00	1.32	1.56	1.80	2.04	2.28	2.60	2.84	3.12
	c	**0.30**	**0.35**	**0.41**	**0.45**	**0.51**	**0.57**	**0.65**	**0.71**	**0.78**
	% c	1.00	1.17	1.37	1.50	1.70	1.90	2.17	2.37	2.60

NVB apical stress, a: apical-third pulpal stress, c: coronal pulpal stress, % NVB: no. of times of stress increase, % a: no. of times of apical-third pulpal-stress increase, % c: no. of times of coronal-stress increase.

**Table 4 ijerph-20-01179-t004:** The Tresca shear-stress average values (KPa) produced by 2.4 N of orthodontic force.

Resorption (mm)		0	1	2	3	4	5	6	7	8
Intrusion 2.4 N	NVB	**6.83**	**7.69**	**8.56**	**9.41**	**10.23**	**12.02**	**13.81**	**15.58**	**17.35**
	% NVB	1.00	1.13	1.25	1.38	1.50	1.76	2.02	2.28	2.54
	a	**0.60**	**0.67**	**0.74**	**0.81**	**0.89**	**1.03**	**1.18**	**1.33**	**1.48**
	% a	1.00	1.12	1.23	1.35	1.48	1.72	1.97	2.22	2.47
	c	**0.60**	**0.67**	**0.74**	**0.81**	**0.89**	**1.03**	**1.18**	**1.33**	**1.48**
	% c	1.00	1.12	1.23	1.35	1.48	1.72	1.97	2.22	2.47
Extrusion 2.4 N	NVB	**6.83**	**7.69**	**8.56**	**9.41**	**10.23**	**12.02**	**13.81**	**15.58**	**17.35**
	% NVB	1.00	1.13	1.25	1.38	1.50	1.76	2.02	2.28	2.54
	a	**0.60**	**0.67**	**0.74**	**0.81**	**0.89**	**1.03**	**1.18**	**1.33**	**1.48**
	% a	1.00	1.12	1.23	1.35	1.48	1.72	1.97	2.22	2.47
	c	**0.60**	**0.67**	**0.74**	**0.81**	**0.89**	**1.03**	**1.18**	**1.33**	**1.48**
	% c	1.00	1.12	1.23	1.35	1.48	1.72	1.97	2.22	2.47
Translation 2.4 N	NVB	**4.46**	**5.08**	**5.72**	**6.36**	**7.02**	**7.87**	**8.70**	**9.53**	**10.34**
	% NVB	1.00	1.14	1.28	1.43	1.57	1.76	1.95	2.14	2.32
	a	**0.43**	**0.49**	**0.54**	**0.59**	**0.64**	**0.73**	**0.82**	**0.91**	**1.00**
	% a	1.00	1.14	1.26	1.37	1.49	1.70	1.91	2.12	2.33
	c	**1.14**	**1.29**	**1.46**	**1.63**	**1.80**	**2.02**	**2.24**	**2.46**	**2.70**
	% c	1.00	1.13	1.28	1.43	1.58	1.77	1.96	2.16	2.37
Rotation 2.4 N	NVB	**6.90**	**9.30**	**11.73**	**14.14**	**16.57**	**18.57**	**20.57**	**22.57**	**24.57**
	% NVB	1.00	1.35	1.70	2.05	2.40	2.69	2.98	3.27	3.56
	a	**0.61**	**0.83**	**1.06**	**1.31**	**1.54**	**1.71**	**1.90**	**2.09**	**2.28**
	% a	1.00	1.36	1.74	2.15	2.52	2.80	3.11	3.43	3.74
	c	**1.16**	**1.72**	**2.05**	**2.49**	**2.91**	**3.27**	**3.62**	**3.97**	**4.31**
	% c	1.00	1.48	1.77	2.15	2.51	2.82	3.12	3.42	3.72
Tipping 2.4 N	NVB	**5.87**	**7.29**	**8.70**	**10.11**	**11.52**	**13.13**	**14.74**	**16.35**	**17.94**
	% NVB	1.00	1.24	1.48	1.72	1.96	2.24	2.51	2.79	3.06
	a	**0.51**	**0.67**	**0.77**	**0.91**	**1.02**	**1.14**	**1.30**	**1.42**	**1.56**
	% a	1.00	1.31	1.51	1.78	2.00	2.24	2.55	2.78	3.06
	c	**0.51**	**0.70**	**0.83**	**0.90**	**1.02**	**1.14**	**1.30**	**1.42**	**1.56**
	% c	1.00	1.37	1.63	1.76	2.00	2.24	2.55	2.78	3.06

NVB apical stress, a: apical- third pulpal stress, c: coronal pulpal stress, % NVB: no. of times of stress increase, % a: no. of times of apical-third pulpal-stress increase, % c: no. of times of coronal-stress increase.

**Table 5 ijerph-20-01179-t005:** The Tresca shear-stress average values (KPa) produced by 4.8 N of orthodontic force.

Resorption (mm)		0	1	2	3	4	5	6	7	8
Intrusion 4.8 N	NVB	**13.66**	**15.38**	**17.11**	**18.81**	**20.47**	**24.05**	**27.62**	**31.16**	**34.17**
	% NVB	1.00	1.13	1.25	1.38	1.50	1.76	2.02	2.28	2.50
	a	**1.19**	**1.34**	**1.48**	**1.62**	**1.77**	**2.06**	**2.36**	**2.66**	**2.95**
	% a	1.00	1.13	1.24	1.36	1.49	1.73	1.98	2.24	2.48
	c	**1.19**	**1.34**	**1.48**	**1.62**	**1.77**	**2.06**	**2.36**	**2.66**	**2.95**
	% c	1.00	1.13	1.24	1.36	1.49	1.73	1.98	2.24	2.48
Extrusion 4.8 N	NVB	**13.66**	**15.38**	**17.11**	**18.81**	**20.47**	**24.05**	**27.62**	**31.16**	**34.17**
	% NVB	1.00	1.13	1.25	1.38	1.50	1.76	2.02	2.28	2.50
	a	**1.19**	**1.34**	**1.48**	**1.62**	**1.77**	**2.06**	**2.36**	**2.66**	**2.95**
	% a	1.00	1.13	1.24	1.36	1.49	1.73	1.98	2.24	2.48
	c	**1.19**	**1.34**	**1.48**	**1.62**	**1.77**	**2.06**	**2.36**	**2.66**	**2.95**
	% c	1.00	1.13	1.24	1.36	1.49	1.73	1.98	2.24	2.48
Translation 4.8 N	NVB	**8.91**	**10.17**	**11.44**	**12.73**	**14.04**	**15.74**	**17.40**	**19.02**	**20.68**
	% NVB	1.00	1.14	1.28	1.43	1.58	1.77	1.95	2.13	2.32
	a	**0.86**	**0.98**	**1.08**	**1.18**	**1.29**	**1.46**	**1.64**	**1.82**	**2.00**
	% a	1.00	1.14	1.26	1.37	1.50	1.70	1.91	2.12	2.33
	c	**2.27**	**2.58**	**2.92**	**3.26**	**3.60**	**4.05**	**4.48**	**4.92**	**5.39**
	% c	1.00	1.14	1.29	1.44	1.59	1.78	1.97	2.17	2.37
Rotation 4.8 N	NVB	**13.79**	**18.60**	**23.45**	**28.28**	**33.14**	**37.14**	**41.14**	**45.13**	**49.14**
	% NVB	1.00	1.35	1.70	2.05	2.40	2.69	2.98	3.27	3.56
	a	**1.23**	**1.66**	**2.12**	**2.62**	**3.08**	**3.42**	**3.81**	**4.18**	**4.55**
	% a	1.00	1.35	1.72	2.13	2.50	2.78	3.10	3.40	3.70
	c	**2.32**	**3.43**	**4.09**	**4.98**	**5.81**	**6.54**	**7.25**	**7.94**	**8.61**
	% c	1.00	1.48	1.76	2.15	2.50	2.82	3.13	3.42	3.71
Tipping 4.8 N	NVB	**11.74**	**14.58**	**17.40**	**20.22**	**23.04**	**26.26**	**29.48**	**32.70**	**35.89**
	% NVB	1.00	1.24	1.48	1.72	1.96	2.24	2.51	2.79	3.06
	a	**1.01**	**1.34**	**1.54**	**1.81**	**2.04**	**2.28**	**2.60**	**2.84**	**3.11**
	% a	1.00	1.33	1.52	1.79	2.02	2.26	2.57	2.81	3.08
	c	**1.21**	**1.40**	**1.65**	**1.80**	**2.04**	**2.28**	**2.60**	**2.84**	**3.11**
	% c	1.00	1.16	1.36	1.49	1.69	1.88	2.15	2.35	2.57

NVB apical stress, a: apical-third pulpal stress, c: coronal pulpal stress, % NVB: no. of times of stress increase, % a: no. of times of apical-third pulpal-stress increase, % c: no. of times of coronal-stress increase.

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
