# Peer review of "Assessment of the Maximum Amount of Orthodontic Force for Dental Pulp and Apical Neuro-Vascular Bundle in Intact and Reduced Periodontium on Bicuspids (Part II)"

_ijerph, 2023, doi:10.3390/ijerph20021179_

Round 1

Reviewer 1 Report

The article presented to me for evaluation deals with a clinically important topic of orthodontic treatment in patients with periodontal diseases and the impact of orthodontic forces on the neurovascular bundle of the tooth.

The introduction is a review of the current literature on the subject. The authors also indicate the possibility of using the finite element method in the assessment of the effects of orthodontic forces on the tooth, in particular its neurovascular bundle.

The goal is clearly defined and the material and methods are well described.

The results of the study are presented clearly in the text as well as in figures and tables. In the discussion, the authors were critical of the results of their research and those of other authors.

The research allowed the authors to draw important conclusions from the clinical point of view, so I believe that the article will be interesting for orthodontists and orthognathic surgeons.

I recommend the article for publication in your valuable journal.

The level of English must be corrected before publication.

Author Response

Department of Cariology, Endodontics and Oral Pathology

Faculty of Dental Medicine

University of Medicine and Pharmacy

Dr. Taylor Tao

Assistant Editor

Special Issue- Advances of Digital Dentistry and Prosthodontics 

                                                                                                                  January 3, 2023

Dear Dr. Taylor Tao,

Thank you very much for your letter dated December 28, 2022, with the comments of the reviewers. We have now carefully considered the comments of the reviewers and amended the paper accordingly. All changes are highlighted in red throughout the manuscript and included also below.

Reply to Reviewer #1:

We agree and we thank the reviewer for his/her time and comments. Appropriate changes in the manuscript have by now been made. Please see below and in the manuscript.

1.Concern of the reviewer:

” The article presented to me for evaluation deals with a clinically important topic of orthodontic treatment in patients with periodontal diseases and the impact of orthodontic forces on the neurovascular bundle of the tooth.

The introduction is a review of the current literature on the subject. The authors also indicate the possibility of using the finite element method in the assessment of the effects of orthodontic forces on the tooth, in particular its neurovascular bundle.

The goal is clearly defined and the material and methods are well described.

The results of the study are presented clearly in the text as well as in figures and tables. In the discussion, the authors were critical of the results of their research and those of other authors.

The research allowed the authors to draw important conclusions from the clinical point of view, so I believe that the article will be interesting for orthodontists and orthognathic surgeons.

I recommend the article for publication in your valuable journal.

The level of English must be corrected before publication.”

Our response:

  • We thank the reviewer for his/her concern and comments. We do hope that our changes are according to the reviewer‘s remarks.

Revised text: Please see the entire manuscript

Reviewer 2 Report

The aim of the study was examine 0.6 N, 1.2 N, 2.4 N and 4.8 N as best orthodontic force for dental pulp and apical NVB in intact and reduced PDL in connection with each individual movement and ischemic, necrotic and resorptive risks. The second aim was to examine if Tresca FEA criterion is more right for in vitro examination of dental pulp and apical NVB.

The topic of the study is interesting and relevant.

In the Introduction the Authors underlined the importance of the research. The aim of the study has been precisely defined.

The methodology applied has been discussed in sufficient details.

The results are quite clearly presented.

In the Discussion the Authors discussed the obtained results and compared with other findings. However, they also should discussed the limitations of the study.

References – Please cite the references according to the Instructions for Authors e.g.

Journal Articles:
1. Author 1, A.B.; Author 2, C.D. Title of the article. 
Abbreviated Journal Name YearVolume, page range.

Minor remarks:

Please use “in vitro” instead of “in vitro” thorough the text e.g. line 108.

Author Response

Department of Cariology, Endodontics and Oral Pathology

Faculty of Dental Medicine

University of Medicine and Pharmacy

Dr. Taylor Tao

Assistant Editor

Special Issue- Advances of Digital Dentistry and Prosthodontics 

                                                                                                                  January 3, 2023

Dear Dr. Taylor Tao,

Thank you very much for your letter dated December 28, 2022, with the comments of the reviewers. We have now carefully considered the comments of the reviewers and amended the paper accordingly. All changes are highlighted in red throughout the manuscript and included also below.

Reply to Reviewer #2:

We agree and we thank the reviewer for his/her time and comments. Appropriate changes in the manuscript have by now been made. Please see below and in the manuscript.

1.Concern of the reviewer:

” The aim of the study was examine 0.6 N, 1.2 N, 2.4 N and 4.8 N as best orthodontic force for dental pulp and apical NVB in intact and reduced PDL in connection with each individual movement and ischemic, necrotic and resorptive risks. The second aim was to examine if Tresca FEA criterion is more right for in vitro examination of dental pulp and apical NVB.

The topic of the study is interesting and relevant.

In the Introduction the Authors underlined the importance of the research. The aim of the study has been precisely defined.

The methodology applied has been discussed in sufficient details.

The results are quite clearly presented.

In the Discussion the Authors discussed the obtained results and compared with other findings. However, they also should discussed the limitations of the study.

References – Please cite the references according to the Instructions for Authors e.g.

Journal Articles:
1. Author 1, A.B.; Author 2, C.D. Title of the article. Abbreviated Journal Name YearVolume, page range.

Minor remarks:

Please use “in vitro” instead of “in vitro” thorough the text e.g. line 108.”

Our response:

  • We thank the reviewer for his/her concern and comments. We do hope that our changes are according to the reviewer‘s remarks.

Revised text: Discussion pg. 14, lines 385-392

“It must be acknowledged as a limitation that the herein simulations were performed with pure orthodontic forces, while in clinical situations (as in those reported in vivo [17-21]) there is of association of them that could lead to the observation that clinically the quantitative stresses might be even smaller than the herein.”

  1. 14, lines 486-497

“Clinically there is an association and combinations of movements, thus, the quantitative stresses that are displayed in the apical NVB, PDL and dental pulp are expected to be lower than the herein. It must be acknowledged as a limit of this method (in addition to those mentioned earlier throughout this study) that the FEA is a mathematical based algorithm method the analysis a clinically extremely complex biomechanical behavior that cannot be entirely reproduced in vitro. Nevertheless, there are no other in vivo or in vitro methods available to investigate these small structures, thus FEA despite all its limits remains the best and only available tool. To gain a better knowledge of the correlations and interrelations between the internal and external orthodontic coronal and radicular resorption, the pulp-PDL-apical NVB complex biomechanical behavior, the amount of applied force and the periodontal breakdown, more FEA simulations employing the suitable failure criteria are in need.”

   Revised text: entire References section pg. 15-17, lines 547-646

  1. “Moga, R. A.; Cosgarea, R.; Buru, S. M.; Chiorean, C. G., Finite element analysis of the dental pulp under orthodontic forces. American journal of orthodontics and dentofacial orthopedics : official publication of the American Association of Orthodontists, its constituent societies, and the American Board of Orthodontics 2019, 155, (4), 543-551.

…”

Revised text: entire manuscript (also pg. 3, line 115)

“The study of stress distribution in PDL, dental pulp and apical NVB is granted only to in vitro simulations due to complexity and anatomical small size of each part, and impossibility of individualized and particularized inquiry in vivo [1, 2, 8].”

Reviewer 3 Report

This paper examined the best orthodotntic force for dental pulp and apical neuro-vascular bundle in the periodontal ligament. The manuscript was well-written and the study methodology is sound. The authors have exhaustively used resources relating to the study. 

If there is one point of improvement, it would be to comprehensively discuss the limitations of this study.

Author Response

Department of Cariology, Endodontics and Oral Pathology

Faculty of Dental Medicine

University of Medicine and Pharmacy

Dr. Taylor Tao

Assistant Editor

Special Issue- Advances of Digital Dentistry and Prosthodontics 

                                                                                                                  January 3, 2023

Dear Dr. Taylor Tao,

Thank you very much for your letter dated December 28, 2022, with the comments of the reviewers. We have now carefully considered the comments of the reviewers and amended the paper accordingly. All changes are highlighted in red throughout the manuscript and included also below.

Reply to Reviewer #3:

We agree and we thank the reviewer for his/her time and comments. Appropriate changes in the manuscript have by now been made. Please see below and in the manuscript.

1.Concern of the reviewer:

” This paper examined the best orthodotntic force for dental pulp and apical neuro-vascular bundle in the periodontal ligament. The manuscript was well-written and the study methodology is sound. The authors have exhaustively used resources relating to the study. 

If there is one point of improvement, it would be to comprehensively discuss the limitations of this study.”

Our response:

  • We thank the reviewer for his/her concern and comments. We do hope that our changes are according to the reviewer‘s remarks.

Revised text: Discussion section pg. 14, lines 385-392

“It must be acknowledged as a limitation that the herein simulations were performed with pure orthodontic forces, while in clinical situations (as in those reported in vivo [17-21]) there is of association of them that could lead to the observation that clinically the quantitative stresses might be even smaller than the herein.”

  1. 14, lines 486-497

“Clinically there is an association and combinations of movements, thus, the quantitative stresses that are displayed in the apical NVB, PDL and dental pulp are expected to be lower than the herein. It must be acknowledged as a limit of this method (in addition to those mentioned earlier throughout this study) that the FEA is a mathematical based algorithm method the analysis a clinically extremely complex biomechanical behavior that cannot be entirely reproduced in vitro. Nevertheless, there are no other in vivo or in vitro methods available to investigate these small structures, thus FEA despite all its limits remains the best and only available tool. To gain a better knowledge of the correlations and interrelations between the internal and external orthodontic coronal and radicular resorption, the pulp-PDL-apical NVB complex biomechanical behavior, the amount of applied force and the periodontal breakdown, more FEA simulations employing the suitable failure criteria are in need.”

Reviewer 4 Report

Dear authors,

thank you for the submission of your paper. It's a very interesting field of research, with few data in literature. 

It could have a wider access to clinician if you agree with our reviews.

Specifically: 

Line 38-41: please specify (with numbers) which are the differences between the “certain level” of pressure in intact and compromised periodontal ligament

Line 65: please specify which volume do you are talking about

Line 107-130: too many sentences without references. Please add references in this section

Materials and method are not clear. How many teeth were analyzed? How do you evaluate the orthodontic force in the reconstructed teeth and how do you? How do you reconstruct missing periodontium? It is only a visual reconstruction? Being visual it could differ from any operators. Considering that there are great differences between different teeth, you have to specify that this study is related exclusively on bicuspids in the title. 

Line 168: characteristics of sample have to be moved in result section

Line 195-196: how do you subject patient to a continue orthodontic force?

Discussion is not well developed: the comparison between results and literature is not simple to understand. It could be addressed in a simpler way, in order to a reach a more fluent read.

Moreover i suggest to extensive review of english form because even if include interesting results, it is difficult to understand in an easy way.

Author Response

Department of Cariology, Endodontics and Oral Pathology

Faculty of Dental Medicine

University of Medicine and Pharmacy

Dr. Taylor Tao

Assistant Editor

Special Issue- Advances of Digital Dentistry and Prosthodontics 

                                                                                                                  January 3, 2023

Dear Dr. Taylor Tao,

Thank you very much for your letter dated December 28, 2022, with the comments of the reviewers. We have now carefully considered the comments of the reviewers and amended the paper accordingly. All changes are highlighted in red throughout the manuscript and included also below.

Reply to Reviewer #4:

We agree and we thank the reviewer for his/her time and comments. Appropriate changes in the manuscript have by now been made. Please see below and in the manuscript.

1.Concern of the reviewer:

” Dear authors,

thank you for the submission of your paper. It's a very interesting field of research, with few data in literature. 

It could have a wider access to clinician if you agree with our reviews.

Specifically: 

Line 38-41: please specify (with numbers) which are the differences between the “certain level” of pressure in intact and compromised periodontal ligament

Line 65: please specify which volume do you are talking about

Line 107-130: too many sentences without references. Please add references in this section

Our response:

  • We thank the reviewer for his/her concern and comments. We do hope that our changes are according to the reviewer‘s remarks.

Revised text: Introduction section, pg:1, lines 41-45

“If in intact periodontium the PDL (with an average thickness of 0.2-0.3 mm) is capable of absorbing and dissipating the orthodontic forces up to a certain amount of force (which is still a debatable issue, with reports of both light and high forces), this ability is seriously affected during periodontal resorption, leading to rapid further tissue loss [1, 2].”

                                          pg:2, lines 68-73

“The apical NVB anatomical structure [with an average of 0.05-0.875mm: 0.002-0.035 mm (40% of volume) for innervation and a similar diameter and volume percentage for the vascular component] is usually classified into three main categories (with various subvariants): thick nerve bundles alongside the blood vessels, thin spiral innervations, and neural bundles that surrounds the blood vessels [3]”

                                            pg:3, lines 114-138

“The study of stress distribution in PDL, dental pulp and apical NVB is granted only to in vitro simulations due to complexity and anatomical small size of each part, and impossibility of individualized and particularized inquiry in vivo [1, 2, 8]. Thus, PDL and pulp studies used either anatomically idealized reconstructions or anatomical simplified CBCT (cone beam computed tomography) based 3D models, while no reconstructions of apical NVB were found [1, 2, 8]. The main issue consists in the small size of apical NVB (well under 1 mm), difficult to be identified and separated in the different shades of gray on the CBCT slice [1, 2, 8]. This process usually can be accurately done only by manual reconstruction performed by an experienced professional [1, 2, 8].

For a proper radiological examination, the CBCT must ensure a voxel size smaller than 0.2 mm (i.e., the average thickness of PDL), and to be able to be used in daily clinical practice (e.g., minimal radiation dose, wide region of interest) [1, 2, 8]. Nevertheless for in vitro examinations, there are micro-CT with an extremely small voxel size but not suited for human clinical use due to high dose of radiations. The radiological data must be provided by daily clinical non-invasive x-ray examinations for ensuring the capture of dis-tinct phases of periodontium under orthodontic treatment [1, 2, 8]. The collected data is reconstructed by an automated imaging software (i.e., grey shades algorithm based) or manually [1, 2, 8]. Due to complexity of tissues, a better anatomical accuracy is provided by the time-consuming manual reconstruction process [1, 2, 8]. The reconstructions are then passed to a FEA (finite elements analysis) software allowing individual simulations and analysis of the entire structure or limited to only one section or part [1, 2, 8].

The provided data is complex and complete, allowing a better understanding of the biomechanical behavior [1, 2, 8]. As a limit of this in vitro method, a FEA simulation cannot completely reproduce a clinical situation, nevertheless, is the only available method to obtain a detailed display of the biomechanical behavior of this type of tissues [1, 2, 8].”

2.Concern of the reviewer:

“Materials and method are not clear. How many teeth were analyzed? How do you evaluate the orthodontic force in the reconstructed teeth and how do you? How do you reconstruct missing periodontium? It is only a visual reconstruction? Being visual it could differ from any operators. Considering that there are great differences between different teeth, you have to specify that this study is related exclusively on bicuspids in the title.” 

Line 168: characteristics of sample have to be moved in result section

Line 195-196: how do you subject patient to a continue orthodontic force?

Our response:

  • We thank the reviewer for his/her concern and comments. We do hope that our changes are according to the reviewer‘s remarks.

Revised text: entire section of Materials and methods, pg:4-5, lines 174-241

     “The herein simulation is the result of a larger stepwise progressive re-search [1, 2, 8, 9] (clinical protocol 158/02.04.2018) aiming to study the maximum amount of orthodontic force safely applied in intact and reduced periodontium without any risks and to identify the suited FEA method/criteria to accurately perform this type of in vitro simulations.

    Our study was performed over nine patients (mean age 29.81 ± 1.45 years, 4 males, informed oral consent) with reduced noninflamed periodontium (i.e., treated chronic periodontitis stage II/III grade B periodontitis, enrolled in supportive therapy). This study initially considered more patients to be included but only nine of them qualified for the eligibility criteria: reduced noninflamed periodontium, complete mandibular arches with no teeth missing, various levels of bone height, indication of orthodontic treatment and availability of follow-up through treatment. The mandibular area (including first and second molar and the two premolars) was investigated by 0.075 mm voxel size CBCT (ProMax 3DS-Planmeca, Finland).

    Based on the CBCT data anatomically correct reconstructions of the structural tissues (i.e., enamel, dentin, dental pulp and its NVB, PDL, cortical and trabecular bone) have been conducted employing the manual image segmentation technique using AMIRA 5.4.0 software (AMIRA, version 5.4.0, Visage Imaging Inc. 300 Brickstone Square, Suite 201 Andover, MA 01810, USA) based on the Hounsfield grey shades units and conducted by a single experienced clinician. Based on the different grey shades the anatomical components were identified and selected individually on each of the DICOM slices of the CBCT. All identified components were than assembled into a single 3D mesh model (i.e., one for each of the nine patients). Thus, nine models having the second premolar (single and two rooted, with different anatomical shapes) and various levels of periodontal breakdown limited mostly to the cervical third of PDL were obtained. The manual image segmentation technique (complex and difficult) was preferred for the supplied anatomical accuracy (the automated reconstruction algorithm refines by simplification committing errors when showing extremely small tissues and almost similar grey shades). In each of the nine models the molars and first premolar were replaced by cortical and trabecular bone, while the missing bone and PDL were reconstructed as accurate as possible in order to obtain nine models with intact periodontium (Figure 1).

    In each model the dentin replaced cementum (due to similarity of its mechanical properties, Table 1), and PDL had a variable average thickness of 0.15-0.225 mm and was reconstructed with its apical NVB. Each of the nine intact periodontium models was the subject of a manual gradual horizontal periodontal breakdown of 1 mm in height (up to 8 mm of loss) obtaining a total number of eighty-one analyzed models (nine models/patient) with the 2nd lower premolar.

     The intact periodontium mesh models had 5.06-6.05 million C3D4 tetrahedral ele-ments, 0.96-1.07 million nodes, and global element size of 0.08-0.116 mm (extremely fine grain mesh subjected to mesh convergence testing). The manual reconstruction technique implied also a limited number of surface anomalies and irregularities naturally present in all models, but with quasi continuity in stressed areas while, nonetheless the internal algorithm-based control processes were passed.

    FEA simulation was performed by employing Tresca failure criterion (share stress) and using ABAQUS 6.13-1 software (Dassault Systèmes Simulia Corp., Stationsplein 8-K, 6221 BT Maastricht, The Netherlands). The orthodontic forces applied over the bracket were 0.6 N (approx. 60 g), 1.2 N (approx. 120 g), 2.4 N (approx. 240 g) and 4.8 N (approx. 480 g), individually for intrusion, extrusion, translation, rotation, and tipping movements (Figure 1), assuming to reproduce as accurately as possible the effects of the movements.

    The physical properties of tissues (boundary conditions-Table 1) were homogeneity, isotropy, linear elasticity, and perfectly bonded interfaces, with no displacements of the base of the models. All models have been subjected to similar boundary conditions, material properties and loading conditions.

     Tresca shear stresses have been found and were displayed qualitatively (i.e., numerically expressed as a color-coded projection) for the dental pulp and its NVB in Figure 2 and 3 (for one of the nine models), while quantitatively (as an average, for all models) in Tables 2-5. The average quantitative values were each time referred to the 16 KPa of physiological MHP, and the risks of external-internal orthodontic coronal and radicular resorption, ischemia-necrosis, and further periodontal loss were individually examined. Stress increase speed was correlated with quantitative stress values for intact periodontium as reference point. The displayed quantitative (Table 2-5) and qualitative results were also associated with our earlier observations [1, 2, 8] employing the Tresca and Von Mises failure criteria for dental pulp and its NVB.”

Revised text: title section

“Assessment of the Maximum Amount of Orthodontic Force for Dental Pulp and Apical Neuro-Vascular Bundle in Intact and Reduced Periodontium on Bicuspids (Part II)”

3.Concern of the reviewer:

“Discussion is not well developed: the comparison between results and literature is not simple to understand. It could be addressed in a simpler way, in order to a reach a more fluent read.

Moreover i suggest to extensive review of english form because even if include interesting results, it is difficult to understand in an easy way.”

Our response:

  • We thank the reviewer for his/her concern and comments. We do hope that our changes are according to the reviewer‘s remarks.

Revised text: entire Discussion section pg:11-14 lines 328-497

4.Concern of the reviewer:

“Moreover i suggest to extensive review of english form because even if include interesting results, it is difficult to understand in an easy way”

Our response:

  • We thank the reviewer for his/her concern and comments. We do hope that our changes are according to the reviewer‘s remarks.

Revised text: entire manuscript

Round 2

Reviewer 4 Report

Dear authors,

I wish to thank you for such modifications you made in order to improve your interesting work, as suggested by reviewers.

I think it is more fluent and precise the first version, following all the correction you made. 

Small issues you still have to address: 

Lines 113-133

I wish to thank you for adding the citation but repeat the same three citations for 11 times could not to be enough. I suggest to you to find more citations to be more appropriate on each sentence. 

Line 177

This line contains n. of patients and mean age: they have to be moved in result section.

Discussion and conclusion have be improved so I think that are fine.

Thank you and good luck.

Author Response

Department of Cariology, Endodontics and Oral Pathology

Faculty of Dental Medicine

University of Medicine and Pharmacy

Dr. Taylor Tao

Assistant Editor

Special Issue- Advances of Digital Dentistry and Prosthodontics 

                                                                                                                  January 4, 2023

Dear Dr. Taylor Tao,

Thank you very much for your letter dated January 4, 2023, with the comments of the reviewers. We have now carefully considered the comments of the reviewers and amended the paper accordingly. All changes are highlighted in red throughout the manuscript and included also below.

Reply to Reviewer #4:

We agree and we thank the reviewer for his/her time and comments. Appropriate changes in the manuscript have by now been made. Please see below and in the manuscript.

1.Concern of the reviewer:

” Dear authors,

I wish to thank you for such modifications you made in order to improve your interesting work, as suggested by reviewers.

I think it is more fluent and precise the first version, following all the correction you made. 

Small issues you still have to address: 

Lines 113-133

I wish to thank you for adding the citation but repeat the same three citations for 11 times could not to be enough. I suggest to you to find more citations to be more appropriate on each sentence. 

Line 177

This line contains n. of patients and mean age: they have to be moved in result section.

Discussion and conclusion have be improved so I think that are fine.

Thank you and good luck

Our response:

  • We thank the reviewer for his/her concern and comments. We do hope that our changes are according to the reviewer‘s remarks.

Revised text: Introduction section, pg:3, lines 113-133

“The study of stress distribution in PDL, dental pulp and apical NVB is granted only to in vitro simulations due to complexity and anatomical small size of each part, and impossibility of individualized and particularized inquiry in vivo [1, 2, 4-8, 13]. Thus, PDL and pulp studies used either anatomically idealized reconstructions or anatomical simplified CBCT (cone beam computed tomography) based 3D models, while no reconstructions of apical NVB were found [1, 2, 8, 9]. The main issue consists in the small size of apical NVB (well under 1 mm), difficult to be identified and separated in the different shades of grey on the CBCT slice [1, 2, 8, 14, 15]. This process usually can be accurately done only by manual reconstruction performed by an experienced professional [1, 2, 8, 9].

For a proper radiological examination, the CBCT must ensure a voxel size smaller than 0.2 mm (i.e., the average thickness of PDL), and to be able to be used in daily clinical practice (e.g., minimal radiation dose, wide region of interest) [1, 2, 4-8]. Nevertheless for in vitro examinations, there are micro-CT with an extremely small voxel size but not suited for human clinical use due to high dose of radiations. The radiological data must be provided by daily clinical non-invasive x-ray examinations for ensuring the capture of distinct phases of periodontium under orthodontic treatment [1, 2, 8, 10, 11]. The collected data is reconstructed by an automated imaging software (i.e., grey shades algorithm based) or manually [1, 2, 4-8]. Due to complexity of tissues, a better anatomical accuracy is provided by the time-consuming manual reconstruction process [1, 2, 8]. The reconstructions are then passed to a FEA (finite elements analysis) software allowing individual simulations and analysis of the entire structure or limited to only one section or part [1, 2, 6-8].

The provided data is complex and complete, allowing a better understanding of the biomechanical behavior [1, 2, 4-8]. As a limit of this in vitro method, a FEA simulation cannot completely reproduce a clinical situation, nevertheless, is the only available method to obtain a detailed display of the biomechanical behavior of this type of tissues [1, 2, 4-8, 13-15].”

2.Concern of the reviewer:

“Line 177

This line contains n. of patients and mean age: they have to be moved in result section.” 

Our response:

  • We thank the reviewer for his/her concern and comments. We do hope that our changes are according to the reviewer‘s remarks.

Revised text: Results section, pg:6, lines 250-252

“The herein FEA analysis was performed over nine patients (mean age 29.81 ± 1.45 years, 4 males, informed oral consent) with reduced noninflamed periodontium, and over a total of eighty-one 3D models (nine models/patient).”
